# OrdinalCLIP: Learning Rank Prompts for Language-Guided Ordinal Regression

**Wanhua Li**[*,1], **Xiaoke Huang**[*,1], **Zheng Zhu**[2], **Yansong Tang**[1], **Xiu Li**[1], **Jie Zhou**[1], **Jiwen Lu**[†,1]

[1]Tsinghua University  [2]PhiGent Robotics

wanhua016@gmail.com  hxk21@mails.tsinghua.edu.cn
zhengzhu@ieee.org  {tang.yansong,li.xiu}@sz.tsinghua.edu.cn
{jzhou, lujiwen}@tsinghua.edu.cn

## Abstract

This paper presents a language-powered paradigm for ordinal regression. Existing methods usually treat each rank as a category and employ a set of weights to learn these concepts. These methods are easy to overfit and usually attain unsatisfactory performance as the learned concepts are mainly derived from the training set. Recent large pre-trained vision-language models like CLIP have shown impressive performance on various visual tasks. In this paper, we propose to learn the rank concepts from the rich semantic CLIP latent space. Specifically, we reformulate this task as an image-language matching problem with a contrastive objective, which regards labels as text and obtains a language prototype from a text encoder for each rank. While prompt engineering for CLIP is extremely time-consuming, we propose OrdinalCLIP, a differentiable prompting method for adapting CLIP for ordinal regression. OrdinalCLIP consists of learnable context tokens and learnable rank embeddings. The learnable rank embeddings are constructed by explicitly modeling numerical continuity, resulting in well-ordered, compact language prototypes in the CLIP space. Once learned, we can only save the language prototypes and discard the huge language model, resulting in zero additional computational overhead compared with the linear head counterpart. Experimental results show that our paradigm achieves competitive performance in general ordinal regression tasks, and gains improvements in few-shot and distribution shift settings for age estimation. The code is available at https://github.com/xk-huang/OrdinalCLIP.

## 1 Introduction

Ordinal regression in computer vision aims to predict a rank number [6] or continue value [21] $y$ for a given data $x$. For example, age estimation aims to estimate the age of a given face image while image aesthetic assessment predicts the aesthetic score for an image. As a fundamental problem, ordinal regression has attracted increasing attention due to its broad applications [18, 28, 25, 15].

A simple solution for ordinal regression is to directly regress a scale value [36]. It is implemented with a single neuron and optimized with $L_1$ or $L_2$ loss. However, it suffers from limited performance. Popular approaches [37, 41, 36, 22] usually discretize the continuous target space into different bins or treat the rank labels as different numbers. Then they regard these discrete values as different categories and perform the classification using a linear layer. These methods are essentially learning *a set of weights to model the concept of order*. There are two main challenges for these methods. First, treating ranks as independent class categories fails to grasp the ordinal property. Second, as the learned concepts are mainly derived from the training set, these approaches are prone to overfit and usually attain unsatisfactory performance.

---

[*] Equal contribution. [†]Corresponding author.

36th Conference on Neural Information Processing Systems (NeurIPS 2022).

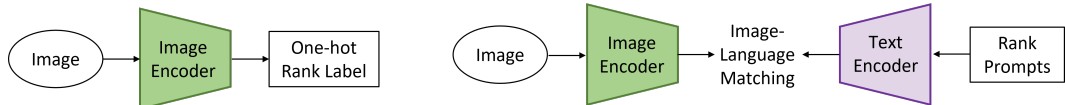

|                                              |                                                              |
|----------------------------------------------|--------------------------------------------------------------|
| (a) Existing Methods for Ordinal Regression  | (b) Language Powered Paradigm for Ordinal Regression (Ours) |

Figure 1: The key idea of our paradigm. Existing methods usually learn the rank concepts from the training data. Instead, we leverage the language priors to learn the rank concepts. We treat each rank category as text and extract the language prototypes with a well-learned text encoder. We align image features into the well-learned language latent space.

Since learning the rank concept from the image domain alone is prone to overfitting, we can leverage multimodal information to alleviate this issue. The human language contains rich semantic information and prior knowledge. We consider simultaneously borrowing the rank concept from the language domain. Specifically, each rank label is not only regarded as a class category, but also linked to a sentence describing the corresponding rank, such as "this person is 23 years old". In this way, our model not only learns the concept of ranks defined on the vision dataset, but also exploits the common knowledge of rank in the language domain. In this paper, we propose a language-powered paradigm for ordinal regression, which *learns a **language prototype** for each rank from a well-defined language latent space*. We aim to associate each rank category with its language concept to alleviate the overfitting issue. We use CLIP [34], which learns an image encoder and a text encoder with large-scale image-text pairs and achieves impressive performance on zero/few-shot knowledge transfer, to leverage the language priors. Specifically, we convert each rank label into text input (prompt) and employ the pre-trained giant text encoder in CLIP to extract language prototypes for all ranks. Then we reformulate ordinal regression as an image-text matching framework, where image embeddings are obtained from a learnable image encoder and a set of language prototypes are served as the text features to be matched. We aim to align the image features into the fixed language space. Since the prototypes are obtained from a fixed language model, we are somehow distilling the language knowledge from the CLIP model. Moreover, the prototypes are constrained in the well-learned language latent space, which is also a kind of regularization leading to stronger generalization.

Recent research [50] has found that the text input (prompt) plays a key role in the performance and choosing the right prompt is extremely time-consuming. Instead of using hard prompts, we propose to learn rank prompts for ordinal regression. The proposed OrdinalCLIP consists of learnable context tokens and learnable rank embeddings. Learnable context tokens are shared across all ranks and used to model the context words. As large language models usually do not fully learn the continuous property of the numbers [43], we also propose to learn rank embeddings by interpolating between several base features, which explicitly model the continuous nature of rank embeddings. The key idea of this paper is shown in Figure 1.

We summarize the main contributions of this paper as follows: (1) We present a new paradigm that learns language prototypes from a fixed language model for ordinal regression. To the best of our knowledge, our method is the first language-powered paradigm for ordinal regression. (2) We introduce learnable rank embeddings, which explicitly model the continuous nature of ranks to better exploit language priors from the text encoder in CLIP. (3) Comprehensive experiments illustrate the effectiveness of the paradigm. OrdinalCLIP achieves competitive performance on three real-world tasks, along with improvements in few-shot and distribution shift settings for age estimation.

## 2   Related Work

**Ordinal Regression in Computer Vision:** Given an input image, ordinal regression in computer vision aims to map the input to a rank or a continuous value. Many popular methods [36, 10] adopt a classification framework. For example, Rothe *et al.* [36] posed facial age estimation as a deep classification problem. Then they computed expected values over the softmax-normalized probabilities for inference. As the simple classification formulation treats different ranks as independent classes, some work [10, 29] attempts to model the relations between ranks. Geng *et al.* [10] proposed label distribution, which regards a sample as an instance associated with a specific label distribution. The mean-variance loss was introduced in [29] to leverage the label ambiguity. Chen *et al.* [3] proposed ranking-CNN with a series of basic CNNs to learn the rank. There are also

some papers [11, 20, 12, 44, 14] modeling ordinal regression from the perspective of probability. Gustafsson *et al.* [11] proposed an energy-based model within a probabilistic regression formulation. Li *et al.* [20] introduced probabilistic ordinal embeddings to model the data uncertainty for regression. DeepGUM [17] used a Gaussian-uniform mixture model to achieve robust performance for outliers. Pan *et al.* [30] presented self-paced deep regression forests to achieve more robust and less biased models. These methods still mainly learn the rank concept from the data, which makes them easy to overfit and leads to sub-optimal performance. Our paradigm aims to learn the concept from the language to alleviate the above issues.

**Language Model:** Recent years have witnessed the great success of pre-training general language representations [4, 2, 35, 33, 5]. ELMo [33] was proposed to extract context-sensitive features. BERT [5] employed masked language models to learn deep bidirectional representations. CLIP [34] proposed to learn representations from image-text pairs. The experimental results show that the CLIP is much more efficient at zero/few-shot transfer, which demonstrates the strong power of language. Encouraged by the excellent performance of CLIP, many CLIP-powered methods have been proposed. Patashnik *et al.* [31] proposed StyleCLIP, which leveraged the power of CLIP models to develop a text-based interface for StyleGAN. Wang *et al.* [45] introduced ActionCLIP for video action recognition. ActionCLIP follows a paradigm "pre-train, prompt and finetune", which achieves strong performance on action recognition tasks. Gao *et al.* [9] proposed the CLIP-Adapter to use the CLIP model on downstream tasks. Zhang *et al.* [48] further proposed Tip-Adapter to enhance the CLIP's few-shot capability without training. Wav2CLIP [47] was proposed to learn robust audio representations by distilling from CLIP models. In this paper, we leverage the language priors from the CLIP model. To better exploit the language priors from CLIP, we further propose differentiable rank prompts to model the continuous property of the numbers. The most relevant works are [50] and [49]. Zhou *et al.* [50] presented a new approach named context optimization. This method learns the context of prompts, which attains significant performance improvements. Zhou [49] also proposed an improved version of [50] by incorporating the visual features into the text features, which improves the generalizability of the model. Note that [50] only optimizes the context embeddings. Thus directly applying it to ordinal regression leads to degraded performance due to the lack of ordinal property. We do not compare with [49], as we want to solely investigate how language priors introduced as language prototypes could help ordinal regression tasks.

# 3 Proposed Approach

## 3.1 Problem Statement

The goal of ordinal regression in computer vision is to estimate the rank number or continue value for a given image. Although we can directly regress this value with a single neuron, a popular baseline is to regard it as a classification task and employ additional constraints trying to make use of the ordinal property. Mathematically, we denote the rank space as $\mathcal{R} = \{r_j | 0 \le j < C\}$, where $r_j$ denotes a rank category and $C$ is the number of ranks categories. During training, we randomly sample a batch of images $\boldsymbol{X} = \{\boldsymbol{x}_i | 0 \le i < B\}$ and the associated labels $\boldsymbol{Y} = \{\boldsymbol{y}_i | 0 \le i < B\}$, where $B$ represents the batch size and $\boldsymbol{y}_i \in \{0, 1\}^C$. The classification baseline treats each rank as a class number. Assuming the rank label for a sample $\boldsymbol{x}_i$ is $r_j$, we construct the one-hot label by setting $y_{i,l} = 1$ if $l = j$, otherwise $y_{i,l} = 0$, where $l$ is an index and $0 \le l < C$.

The images are first sent to an image encoder to extract image features $\boldsymbol{f} = \{\boldsymbol{f}_0, \boldsymbol{f}_1, \dots, \boldsymbol{f}_{B-1}\}$, where $\boldsymbol{f}_i \in \mathbb{R}^d$ is the extracted features of image $\boldsymbol{x}_i$. We use $d$ to denote the feature dimension. Then the image features are sent to a linear layer with a set of parameters $\boldsymbol{W} = [\boldsymbol{w}_0, \boldsymbol{w}_1, \dots, \boldsymbol{w}_{C-1}]^\top \in \mathbb{R}^{C \times d}$ and $\boldsymbol{b} = [b_0, b_1, \dots, b_{C-1}] \in \mathbb{R}^C$:

$$l_{i,j} = \boldsymbol{w}_j^\top \boldsymbol{f}_i + b_j \,, \tag{1}$$

where $l_{i,j}$ is the predicted logit for rank $r_j$ of image $\boldsymbol{x}_i$. Then the model is optimized with a cross-entropy loss:

$$\mathcal{L}_{cls} = \frac{1}{B} \sum_{i=0}^{B-1} \sum_{j=0}^{C-1} -y_{i,j} \log \frac{\exp(l_{i,j})}{\sum_{j=0}^{C-1} \exp(l_{i,j})} \,. \tag{2}$$

Many existing methods develop additional technologies based on the classification formulation, such as label distribution [10, 8, 7], mean-variance loss [29], and so on. However, these methods simply

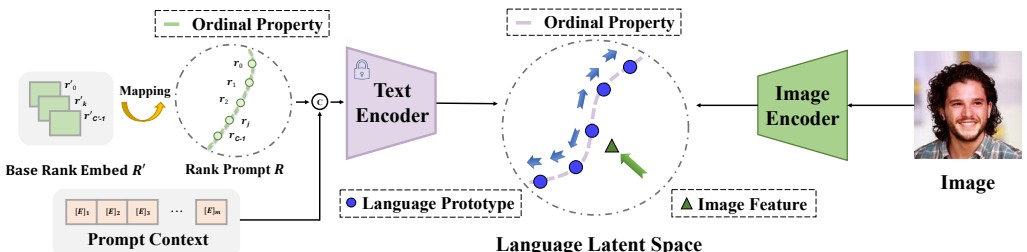

Figure 2: The framework of OrdinalCLIP. We regard rank categories as text and employ a language model to leverage the language priors. For each rank, we concatenate its word embedding and learnable prompt context. Then they are sent to a language model to extract the corresponding language prototype. To preserve the ordinal property of language prototypes, we explicitly construct the ordinal rank embeddings that are interpolated from several base rank embeddings. We found the ordinality of the rank embeddings can be implicitly propagated toward the language prototypes.

treat each rank as a number and learn the rank concept mainly from the training data leading to the overfitting problem along with unsatisfactory performance. In this paper, we propose a language-powered paradigm, which learns the rank concept from language to address the above issues.

### 3.2 Language Prototypes

We believe that natural language contains rich prior knowledge. To preserve ordinality, we propose to link the learned rank category to its language concept. To leverage the language power, we employ the text encoder of CLIP [34] as the language model. CLIP follows a vision-language pre-training framework and learns representations from image-text pairs, resulting in a joint vision-language latent space. As the CLIP model shows strong performance in downstream tasks, we implement our method with the CLIP-powered text encoder.

We reformulate the ordinal regression task as an image-language matching problem, which consists of a text encoder and an image encoder. The text encoder is used to extract language features with the fixed CLIP language model. As the parameters of the text encoder are fixed, the text is mapped to a well-learned language latent space, which encodes rich language priors.

To leverage the language priors with the text encoder, we treat the rank categories as words. Then we obtain a set of words for the rank categories $\{[r_0], [r_1], \ldots, [r_{C-1}]\}$. To obtain the language embedding for each rank, we construct a sentence for these ranks and send the entire sentence to the text encoder. For example, we can construct a sentence of *"a person at the age of $[r_j]$"* for the rank $[r_j]$. Then each word of this sentence is mapped to a 512-D word embedding vector.[1] All the word embeddings are sent to the text encoder to obtain a text embedding. In this way, each rank category is mapped to a text embedding in the language latent space. We treat the mapped embedding as the prototype of its corresponding rank category. As these prototypes lie on the well-learned language manifold and encode the language concept, we name them as **language prototypes**. Formally, we use $\{\boldsymbol{p}_j \in \mathbb{R}^d | 0 \leq j < C-1\}$ to represent the $\ell_2$ normalized language prototypes, where $\boldsymbol{p}_i$ corresponds to the language prototype of the rank $r_j$ and $d$ is the feature dimension.

The input images are processed by an image encoder to extract image features. We can employ any popular vision backbones as the image encoder. While the parameters of the text encoder are fixed, we train the entire image encoder to align image features to the language latent space. For a batch of images $\boldsymbol{X}$, we denote the $\ell_2$ normalized image features as $\boldsymbol{I} = \{\boldsymbol{I}_i \in \mathbb{R}^d | 0 \leq i < B-1\}$, where $d$ is the feature dimension. As an image-language matching framework, we calculate the similarity scores between the two modalities through the inner product and obtain a similarity matrix $\boldsymbol{A} = [a_{i,j}]$:

$$a_{i,j} = \boldsymbol{I}_i \cdot \boldsymbol{p}_j^\top, 0 \leq i < B-1, 0 \leq j < C-1. \tag{3}$$

We use image-text contrastive loss to optimize the network, which consists of an image-to-text loss and a text-to-image loss following the practice of CLIP [34]. For image-to-text loss, we first

---

[1]Or several 512-D sub-word embeddings due to the usage of BPE [39]

normalize the rows of the matrix $\boldsymbol{A}$ using a softmax layer to obtain the matrix $\boldsymbol{A}' = [a'_{i,j}]$:

$$a'_{i,j} = \frac{\exp(a_{i,j}/T)}{\sum_{j=0}^{C-1} \exp(a_{i,j}/T)} \, , \tag{4}$$

where $T$ is the temperature parameter. We minimize the discrepancy between the matrix $\boldsymbol{A}'$ and label $\boldsymbol{Y}$ using KL divergence. For text-to-image loss, there may be zero or multiple hits. Therefore we construct the label matrix $\boldsymbol{Y}''$ by normalizing the non-zero columns of $\boldsymbol{Y}$ [45]. Besides, we obtain a column-normalized matrix $\boldsymbol{A}'' = [a''_{i,j}]$:

$$a''_{i,j} = \frac{\exp(a_{i,j}/T)}{\sum_{i=0}^{B-1} \exp(a_{i,j}/T)} \, . \tag{5}$$

We also employ a KL divergence to minimize the discrepancy between the matrix $\boldsymbol{A}''$ and the label matrix $\boldsymbol{Y}''$. In the end, we have the image-text contrastive loss:

$$\mathcal{L} = \frac{1}{2} [ \frac{1}{B} \sum_{i=0}^{B-1} KL(\boldsymbol{Y}_{i,.}, \boldsymbol{A}'_{i,.}) + \frac{1}{C} \sum_{j=0}^{C-1} KL(\boldsymbol{Y}''_{.,j}, \boldsymbol{A}''_{.,j})] \, . \tag{6}$$

### 3.3 Learning Rank Prompts

As illustrated in [34, 50], different text input, which is known as prompt, may give very different performances. Instead of designing the prompt context, we directly learn $m$ word embeddings $[E]_0, [E]_1, \ldots, [E]_{m-1}$ to serve as the prompt context, which is concatenated with the word embeddings of ranks and is sent to the text encoder.

To further strengthen the ordinality of our model and improve its performance, we propose to learn the rank embeddings to preserve the order of the language prototypes in the language latent space. The reason why we cannot directly apply CLIP to regression is that the human-built prompts from the rank labels lack ordinality (see Figure 3), which leads to degraded performance. Inspired by the arithmetic property in the word embeddings space [26, 32, 1], whose typical example is the following embedding arithmetic equation: "King − Man + Woman = Queen", we choose to maintain the order of rank embeddings to preserve the order of the language prototypes. By explicitly constructing continuous rank embeddings, we can also improve the ordinality of the language prototypes (also see Figure 3). We regard this ordinality as the natural outcome of continuity.

We derive the ordinal rank embeddings from a set of base rank embeddings via interpolation. Suppose we have a set of base rank embeddings $\boldsymbol{R}' = [\boldsymbol{r}'_0, \boldsymbol{r}'_1, \ldots, \boldsymbol{r}'_{C'-1}]$, where $C'$ denotes the number of the embeddings, and $\boldsymbol{R}' \in \mathbb{R}^{512 \times C'}$. Note that in practice, $C'$ should be a relatively small number compared with $C$ (i.e., $C' << C$). Then the j-th rank embedding could be obtained by:

$$\boldsymbol{r}_j = \sum_{k=0}^{C'-1} w'_{j,k} \cdot \boldsymbol{r}'_k \, . \tag{7}$$

where $0 \le k < C'$, and $\boldsymbol{W}' = [w'_{j,k}] \in \mathbb{R}^{C \times C'}$ is the interpolation weights:

$$w'_{j,k} = \frac{I(w_{j,k})}{\sum_{k=0}^{C'-1} I(w_{j,k})} \, , w_{j,k} = |j - \frac{C-1}{C'-1} \cdot k| \, . \tag{8}$$

where $I(\cdot)$ is the interpolation function, and $|\cdot|$ outputs absolute value. We simply propose two interpolation approaches, which are *linear interpolation*: $I(w_{j,k}) = 1 - \frac{w_{j,k}}{C-1}$ and *inverse-proportion interpolation*: $I(w_{j,k}) = \frac{1}{w_{j,k}+\epsilon}$. We set $\epsilon$ to 1e-5 for numerical stability. Figure 2 presents the details of the rank prompt of OrdinalCLIP.

Although our method uses a huge language model to exploit the language concepts for training. We can only employ the learned language prototypes during testing. In this way, our method does not bring any additional computational and storage overhead compared with the linear head alternative.

Table 1: Results on MORPH II.

| Methods | MAE | Year |
|---|---|---|
| AGEn [42] | 2.52 | 2018 |
| BridgeNet [21] | 2.38 | 2019 |
| AVDL [46] | 2.37 | 2020 |
| POE [20] | 2.35 | 2021 |
| CNN Baseline | 2.63 | - |
| Zero-shot CLIP [34] | 14.45 | 2021 |
| CoOp [50] | 2.39 | 2021 |
| OrdinalCLIP | **2.32** | - |

Table 2: Results on the Adience face dataset.

| Methods | Accuracy(%) | MAE |
|---|---|---|
| CNNPOR [23] | $57.4 \pm 5.8$ | $0.55 \pm 0.08$ |
| GP-DNNOR [24] | $57.4 \pm 5.5$ | $0.54 \pm 0.07$ |
| SORD [6] | $59.6 \pm 3.6$ | $0.49 \pm 0.05$ |
| POE [20] | $60.5 \pm 4.4$ | $\mathbf{0.47 \pm 0.06}$ |
| CNN Baseline | $56.0 \pm 5.9$ | $0.56 \pm 0.09$ |
| Zero-shot CLIP [34] | $25.4 \pm 3.5$ | $1.50 \pm 0.20$ |
| CoOp [50] | $60.6 \pm 5.5$ | $0.50 \pm 0.08$ |
| OrdinalCLIP | $\mathbf{61.2 \pm 4.2}$ | $\mathbf{0.47 \pm 0.06}$ |

Table 3: Results of different interpolation types with different numbers of base rank embeddings on the MORPH II dataset. Report MAE for each combination of settings.

| # Base Ranks | 2 | 3 | 4 | 5 | 6 | 7 | 8 | 9 |
|---|---|---|---|---|---|---|---|---|
| Linear | 2.38 | **2.32** | 2.38 | 2.36 | 2.39 | 2.38 | 2.40 | 2.41 |
| Inv. Prop | 2.39 | 2.38 | 2.39 | 2.40 | 2.39 | **2.35** | 2.38 | 2.38 |

# 4 Experiments

## 4.1 Age Estimation

**Datasets:** The task of facial age estimation is to predict the age of a given facial image. We test our method on the widely-used MORPH II [13] dataset and Adience [19] dataset. Morph II contains 55,134 portraits from 13,618 individuals. Each portrait image is labeled with an age value from 16 to 77. Popular evaluation protocol [36, 40, 21, 20] is adopted in our experiments, which only contains 5,492 images of Caucasian descent to remove the cross-race interference. For general regression, we select 80% of the images for training, and others for testing. Note the long-tailed distribution in the dataset. Besides, the distribution of the test set constructed by random sampling is consistent with that of the training data. Adience contains 26,580 Flickr photos from 2,284 subjects. The label is annotated with eight groups, which are 0-2, 4-6, 8-13, 15-20, 25-32, 38-43, 48-53, and over 60-year-old. We adopt the same five-fold cross-validation protocol as used in [23]. We adopted mean average error (MAE) to measure the absolute differences between the ground truth labels and the predicted ones. In addition, classification accuracy is adopted for Adience. For the detailed experimental settings please refer to Appendix.

**Comparison with State-of-the-art methods:** We report the results in Table 1. The state-of-the-art methods are usually designed especially for age estimation [46, 42]. DRFs [40] learned a Gaussian mixture model by extending the deep neural decision forests [16] for regression. AVDL [46] leveraged a meta-learning framework to learn an adaptive variance for label distribution. POE [20] proposed a general regression framework with uncertainty modeling and achieved superior performance. Compared with the advanced approaches, zero-shot CLIP only obtains an MAE of 14.45, implying the poor modeling of numerical and ordinal concepts. CoOp scores an MAE of 2.39, illustrating the effectiveness of the language priors. OrdinalCLIP achieves a superior MAE of 2.32 without bells and whistles, resulting from the explicit ordinal constraint. We found similar results in the Adience [23] dataset. Specifically, CoOp attains 60.6% for regression accuracy, which is better than the previous best. Our OrdinalCLIP outperforms previous state-of-the-arts with an accuracy of 61.2% and is on par with the previous best method under the MAE metric of 0.47.

**Parameter Analysis:** We first analyze the influence of different numbers of base rank embeddings and different interpolation types. With all parameter combinations, the models can all achieve comparable MAEs on MORPH II(Table 3). When varying the number of base rank embeddings, the results improve slightly until a certain sweet point. The best MAE of 2.32 is reported with 3 base ranks. For *inverse-proportion* one, the best MAE of 2.35 is attained with 7 base ranks. However, its overall performance is sub-optimal compared with the *linear* interpolation one. In the following experiments, we chose *linear* interpolation and 3 base ranks as the default hyperparameters.

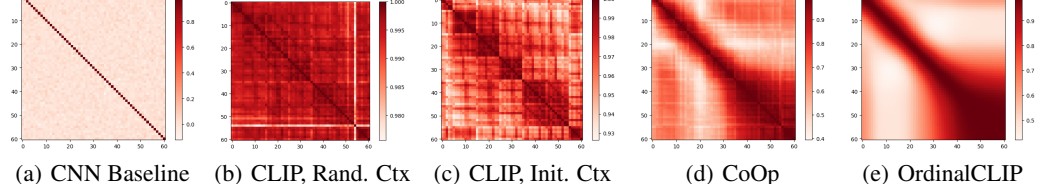

|     (a) CNN Baseline     |     (b) CLIP, Rand. Ctx     |     (c) CLIP, Init. Ctx     |     (d) CoOp     |     (e) OrdinalCLIP     |

Figure 3: The similarity matrices of language prototypes for different rank prompts. The redder, the more similar the pair of language prototypes. The percentages of prototype pairs that obey the ordinality are: 49.13%, 49.71%, 54.84%, 59.92%, and **65.94%**, respectively.

Table 4: Ablation experiments on the MORPH II dataset.

| Choice | Regression Methods | | | | | | | | | | |
|---|---|---|---|---|---|---|---|---|---|---|---|
| | CoOp [50] | | | | | | OrdinalCLIP | | | | |
| Tune Rank | ✓ | | ✓ | ✓ | | ✓ | ✓ | | ✓ | ✓ | | ✓ |
| Tune Ctx | | ✓ | ✓ | | ✓ | ✓ | | ✓ | ✓ | | ✓ | ✓ |
| Init Ctx | | | | ✓ | ✓ | ✓ | | | | ✓ | ✓ | ✓ |
| MAE | 2.58 | 2.44 | 2.39 | 2.41 | 2.39 | 2.39 | 2.38 | 2.37 | 2.36 | 2.34 | 2.34 | **2.32** |

**Ordinality of Learned Language Prototypes:** We validate the ordinality of the rank embeddings by measuring the similarity of their corresponding language prototypes. OrdinalCLIP makes predictions by measuring the similarity between the normalized image embedding and each normalized language prototype in the CLIP latent space. Ideally, the structure of the normalized language prototypes should preserve the ordinality - "lie on a line" in the CLIP latent space. This is aligned with human perception and judgment while facing ordinal regression tasks. We compute the similarity matrices followed by Equation 4 with both inputs being language prototypes. The derived matrices are then max-normalized by their maximum similarity. The ordinality score is the percentage of language prototype pairs that obey the ordinal property[2].

Figure 3 visualizes the ordinality under various situations. The CNN baseline (Figure 3(a)) has the least ordinality score of 49.13%; Half of the language prototype pairs violate the ordinal property and all the prototypes are distributed quite sparsely across the latent space. Figure 3(b) and Figure 3(c) are results computed directly from the CLIP model. Though the initialized context embeddings help to improve ordinality (from 49.71% to 54.84%), almost half of the language prototype pairs violate the ordinal properties. [34] found that CLIP models give degraded predictions facing abstract concepts like numbers. We attribute this to the loss of ordinality in the CLIP latent. Figure 3(d) shows that trained context embeddings from CoOp could further improve the ordinal property with an ordinality score of 59.92%. However, the learned language prototypes are still "twisted" in the CLIP space, leading to degraded predictions. Our OrdinalCLIP (Figure 3(e)) learns smoother language prototypes with a boosted ordinality score of 65.94%, which validates the effectiveness of the learned base rank and their interpolation. Note that the similarity matrices reflect the long-tail property of the dataset. Due to the insufficient samples for the older people, there is a "red blob" in the right-down corner. CLIP and CoOp cannot push the "older" labels away and leave them closer to the well-learned "younger" labels, which may lead to corrupted predictions. Our OrdinalCLIP reinforces the ordinality and as a result, outperforms the above two methods.

**Ablation Study:** To validate the effectiveness of OrdinalCLIP, we compare OrdinalCLIP with preceded CoOp [50] using different optimization and prompt context initialization strategies. There are four major observations in Table 4.

First, for both CoOP and OrdinalCLIP, solely optimizing either rank embeddings or context embeddings leads to sub-optimal performance, yet training them together results in optimal results. Note that CoOp only optimizes the context embeddings while leaving the rank embeddings untouched as they are initialized with the rank labels. However, after jointly training with the rank embeddings, the performance has been boosted from 2.44 to 2.39. For OrdinalCLIP, that is from 2.37 to 2.36 even with random base rank embeddings, resulting from the explicit ordinality from our method.

---

[2]$i.e., \left(\sum_{i=0,j=0,i<=j}^{N-2,N-2} \mathbf{1}\{a'_{i,j} > a'_{i,j+1}\}\right)/((N-1) \times N/2).$

Table 5: We report the MAE results under few-shot settings on the MOPRH II dataset.

| # Shots | 1 | 2 | 4 | 8 | 16 | 32 | 64 |
|---|---|---|---|---|---|---|---|
| CNN Baseline | 8.04 | 6.00 | 4.66 | 3.91 | 3.57 | 3.06 | 2.84 |
| CoOp [50] | 5.09 | 4.50 | 3.81 | 3.57 | 3.23 | 2.87 | 2.61 |
| OrdinalCLIP | 4.94 | 4.36 | 3.55 | 3.31 | 3.07 | 2.76 | 2.57 |

Table 6: The MAE results under the distribution shift setting on the MOPRH II. "re cls" denotes the number of reduced classes, and "re smp" means the percentage of reduced sampled in one class.

| re cls - re smp | 10-80 | 10-90 | 20-80 | 20-90 | 30-80 | 30-90 | 40-80 | 40-90 |
|---|---|---|---|---|---|---|---|---|
| CNN Baseline | 2.88 | 2.98 | 3.46 | 3.68 | 3.19 | 4.06 | 3.28 | 3.65 |
| CoOp [50] | 2.71 | 2.85 | 2.98 | 3.51 | 3.06 | 3.36 | 2.99 | 3.30 |
| OrdinalCLIP | 2.61 | 2.67 | 2.77 | 3.06 | 2.86 | 3.21 | 2.84 | 3.12 |

Second, the initialized rank embeddings are prone to be degraded, which is another observation validating that the CLIP latent space struggles to represent abstract concepts like numbers [34]. When only optimizing rank embeddings, CoOp obtains an MAE of 2.41 with initialized context prompts. In contrast, our OrdinalCLIP could achieve a better MAE of 2.38 even with all the context and base rank embeddings initialized randomly. We attribute this to the explicit ordinal modeling – the interpolation between the base ranks.

Third, initializing prompts with task-specific context could boost performance, no matter whether it is CoOp or OrdinalCLIP, or which kind of embeddings are optimized. For example, our OrdinalCLIP could achieve the best MAE of 2.32 when training both embeddings and initializing the context.

Last, OrdinalCLIP outperforms CoOp under all settings in the ablation, which further validates the effectiveness of the proposed learned base rank embeddings and their interpolation.

**Few-Shot Learning:** We validate the generalization ability of OrdinalCLIP for the few-shot regression learning. The full dataset is split into 80% for training and 20% for testing. We use the entire test set for validation and only choose 1/2/4/8/16/32/64 samples in the training set from each class of the labels for training. We report the results in Table 5. We observe the language models consistently outperform the baseline by a large margin. By incorporating the language prior, the MAE is improved by 2.95 compared with baseline under 1 shot setting for CoOp; Our OrdinalCLIP further improves the MAE to 4.94. For other shot settings, we also obverse the same performance gains, which shows the effectiveness of the proposed learned rank embeddings.

**Distribution Shift:** Another generalization experiment about data distribution shifts is conducted on the MORPH II database. We also use the entire test set of the general regression setting. For the training set, we randomly choose several rank labels, and then randomly discard a large portion of samples in those classes. Table 6 shows the comparison results. We observe that by incorporating language prior, CoOp improves performance by a large margin compared with the baseline. While OrdinalCLIP boosts performance furthermore, illustrating our OrdinalCLIP can better resist the distribution shift of dataset by the explicit ordinal constraint. Specifically, with 40 rank labels selected and 90% of their samples discarded (most severe distribution shift setting), CoOp obtains a 0.35 improvement of MAE compared with baseline, while OrdinalCLIP further attains a performance gain of 0.18.

## 4.2 Image Aesthetics Assessment

**Dataset.** The image aesthetic dataset [38] contains 13,929 available Flickr photos of four categories, i.e, nature, animal, urban, and people. Five absolute rating scales are employed to evaluate the aesthetics quality: "unacceptable", "flawed", "ordinary", "professional", and "exceptional". Each image is judged by at least five different examiners and uses the median rank as ground truth. The whole dataset is randomly split into three non-overlapped subsets following [38]. Five-fold cross-validation was used for fair comparisons. We reported the mean values for both MAE and classification accuracy. We detail the experimental settings in Appendix.

Table 7: Quantitative Results on the Image Aesthetics dataset. We report accuracy and MAE.

| Methods | Accuracy(%) - higher is better | | | | | MAE - lower is better | | | | |
|---|---|---|---|---|---|---|---|---|---|---|
| | Nature | Animal | Urban | People | Overall | Nature | Animal | Urban | People | Overall |
| CNNPOR [23] | 71.86 | 69.32 | 69.09 | 69.94 | 70.05 | 0.294 | 0.322 | 0.325 | 0.321 | 0.316 |
| SORD [6] | 73.59 | 70.29 | **73.25** | 70.59 | 72.03 | 0.271 | 0.308 | **0.276** | 0.309 | 0.290 |
| POE [20] | 73.62 | 71.14 | 72.78 | 72.22 | 72.44 | **0.273** | 0.299 | 0.281 | 0.293 | 0.287 |
| CNN Baseline | 72.76 | 70.44 | 70.12 | 70.80 | 71.03 | 0.338 | 0.381 | 0.364 | 0.377 | 0.365 |
| Zero-shot CLIP [34] | 63.33 | 36.78 | 49.84 | 24.79 | 43.69 | 0.438 | 0.737 | 0.683 | 1.063 | 0.730 |
| CoOp [50] | 72.74 | 71.46 | 72.14 | 69.34 | 71.42 | 0.285 | 0.298 | 0.294 | 0.330 | 0.302 |
| OrdinalCLIP | **73.65** | **72.85** | 73.20 | **72.50** | **73.05** | **0.273** | **0.279** | 0.277 | **0.291** | **0.280** |

**Results:** Table 7 shows the comparison with the previous SOTA methods. Language priors assist the model to achieve better performance, while zero-shot CLIP still struggles to distinguish ordinal concepts. Our OrdinalCLIP attains an overall MAE of 0.280 and overall accuracy of 72.58%, outperforming the previous best method. The results across all nominal categories either outperform or are on par with previous state-of-the-art. Consistent improvements are observed across all categories compared with baseline and CoOp, showing the effectiveness of learned rank prompts with the ordinal property.

## 4.3 Historical Image Dating

**Dataset:** The historical image dating dataset [27] is a benchmark for automatically predicting the decade of the historical colored image. There are five-decade categories in the database from the 1930s to the 1970s, where each category contains 265 images. For the general regression setting, the data of each category is divided into three parts: 210 for training, 5 for validation, and the rest for testing. Ten-fold cross-validation is adopted following [23, 27]. We report the mean and standard deviation for both MAE and accuracy metrics. Refer to Appendix for the detailed experimental settings.

Table 8: Results on the Historical Image Dating.

| Methods | Acc (%) | MAE |
|---|---|---|
| CNNPOR [23] | 50.12 ± 2.65 | 0.82 ± 0.05 |
| GP-DNNOR [24] | 46.60 ± 2.98 | 0.76 ± 0.05 |
| POE [20] | 54.68 ± 3.21 | 0.67 ± 0.04 |
| CNN Baseline | 42.56 ± 3.04 | 0.80 ± 0.03 |
| Zero-shot CLIP [34] | 26.08 ± 0.56 | 1.48 ± 0.03 |
| CoOp [50] | 51.90 ± 2.60 | 0.76 ± 0.06 |
| OrdinalCLIP | **56.44 ± 1.66** | **0.67 ± 0.03** |

**Results:** We compare our OrdinalCLIP with other state-of-the-art models using the full dataset. Table 8 demonstrates that CoOp improves the average accuracy by 9.34% compared with the baseline method, which again shows the importance of exploiting language priors. Zero-shot CLIP is still suffering from degraded results due to pool ordinal transferability. Our OrdinalCLIP further improves the accuracy by 4.45%. Compared with the previous best, OrdinalCLIP attains a competitive MAE of 0.67 and achieves a new state-of-the-art accuracy of 56.44% with only half of the standard deviation, which validates the effectiveness of the proposed learned rank embeddings.

## 5 Discussions and Conclusions

OrdinalCLIP is a general approach for ordinal regression. It can be deployed on many vision tasks such as age estimation. These tasks may be used for surveillance without user permission, which could lead to privacy issues. Therefore, strict technical controls may be needed. Since our method directly employs existing language models, the bias in the language models will be also inherited by our method. How to alleviate the bias in the language model still needs further research.

In this paper, we have presented the language-powered paradigm for ordinal regression. Existing ordinal regression methods usually suffer from the overfitting problem and lack of ordinality within the latents, as they mainly learn the rank concepts from the training data. We propose OrdinalCLIP, which associates each rank category with its language concept derived from the CLIP text encoder. To leverage the language priors, each rank is mapped to a corresponding language prototype. We further

propose learnable rank prompts to explicit learning ordinal rank embeddings to preserve the order of the language prototypes in the language latent space. Extensive experimental results from three tasks including age estimation, historical image dating, and image aesthetics assessment show that our OrdinalCLIP attains very competitive performance in general ordinal regression tasks. OrdinalCLIP also gains improvements in few-shot and distribution shift settings for age estimation.

## Acknowledgment

This work was supported in part by the National Key Research and Development Program of China under Grant 2017YFA0700802, in part by the National Natural Science Foundation of China under Grant 62125603 and Grant U1813218, in part by a grant from the Beijing Academy of Artificial Intelligence (BAAI).

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
