# A Experimental Settings

All experiments were conducted on a single NVIDIA RTX 3090 GPU.

**MORPH II & Adience:** For the image encoder, we followed the common practices in [9, 10, 5, 4] and employed a VGG-16 [11] network for MORPH II, which was pre-trained on the large-scale IMDB-WIKI dataset [9]. Meanwhile, we adopted an ImageNet pre-trained VGG-16 for Adience dataset following [1, 7]. The extracted image features were then projected to the CLIP latent space with an additional Fully-Connected (FC) layer. For the text encoder, we chose the open-sourced CLIP [1] implementation. We utilized the text encoder which was pre-trained with the modified ResNet50 [2, 8] image encoder.

The obtained text features were also projected into the CLIP latent space via an FC layer. The dimension of the CLIP space was set to 1024, following the official release from OpenAI. To validate the non-trivial choice of bringing ordinal regression into CLIP latent space, we also constructed another CNN baseline, with a set of learnable embeddings to replace the true text embeddings, We found that initializing the prompt context gave better performance (Table 4). The prompt context was initialized to *"age estimation: the age of the person is {age}"*, where *{age}* is the label of the image. We used the RAdam optimizer [6] with a learning rate of 1e-4 for the image encoder, the learnable context embeddings, and the base rank embeddings for fast convergence since there are more rank categories in MOPRH II. As for Adience, we took the Adam [3] Optimizer. The model was trained for 50 epochs and the learning rate was decayed with a factor of 0.1 at epoch 30. To prevent overfitting, we performed simple data augmentations following [7]. The training images were first resized to $256 \times 256$ and then cropped to $224 \times 224$ randomly, followed by a randomly horizontal flip operation. The test images followed the same process except that the center cropping was used. We adopted mean average error (MAE) to measure the absolute differences between the ground truth labels and the predicted ones. Besides, the classification accuracy is adopted for Adience.

**Image Aesthetics Assessment** An ImageNet pre-trained VGG-16 was used as the image encoder. The initial learning rates for the image encoder, the context embeddings, and the base rank embeddings were set to 1e-4, and the learning rate for the last layer of the image encoder was set to 1e-3 for faster convergence. The above learning rates decayed by a factor of 0.1 at epochs 25 and 35, respectively. We trained the model for 50 epochs with Adam [3] and applied the data augmentation mentioned above. The initial prompt was *"aesthetics assessment: the quality of the photo is {rank}"*. We reported mean and standard deviation for both MAE and classification accuracy.

**Historical Image Dating:** We used an ImageNet pre-trained VGG-16 as the image encoder. The model was trained for 50 epochs with Adam [3] optimizer. The initial learning rates for the image encoder, the context embeddings, and the base rank embeddings were set to 1e-4, and the learning rate for the last layer of the image encoder was set to 1e-3 for faster convergence. The above learning rate decayed by a factor of 0.1 at epochs 25 and 35, respectively. The initial prompt was *"historical dating: a photo taken in the {label}"*. We pre-processed the training data by first resizing them into $256 \times 256$, then randomly cropped them into $224 \times 224$. We took random horizontal flipping as data augmentation. Both classification accuracy and MAE were reported on this dataset.

# B Dataset Details

In this paper, we only use the existing publicly available data. For those datasets containing personal data, we actively contact the dataset creators and consult the progress of IRB.

Figure 1 shows the original and shifted distributions of the MORPH II dataset. We list several randomly selected samples from the historical image dating dataset (Figure 3) and image aesthetics assessment dataset (Figure 2) to give better descriptions of the above two tasks.

# C More Results of Ordinality of Learned Language Prototypes

In this section, we present comprehensive experimental results of ordinality [2] of language prototypes on the MORPH II dataset.

---

[1] https://github.com/openai/CLIP. MIT License.

[2] *i.e.,* $(\sum_{i=0,j=0,i<=j}^{N-2,N-2} \mathbf{1}\{a'_{i,j} > a'_{i,j+1}\})/((N-1) \times N/2)$.

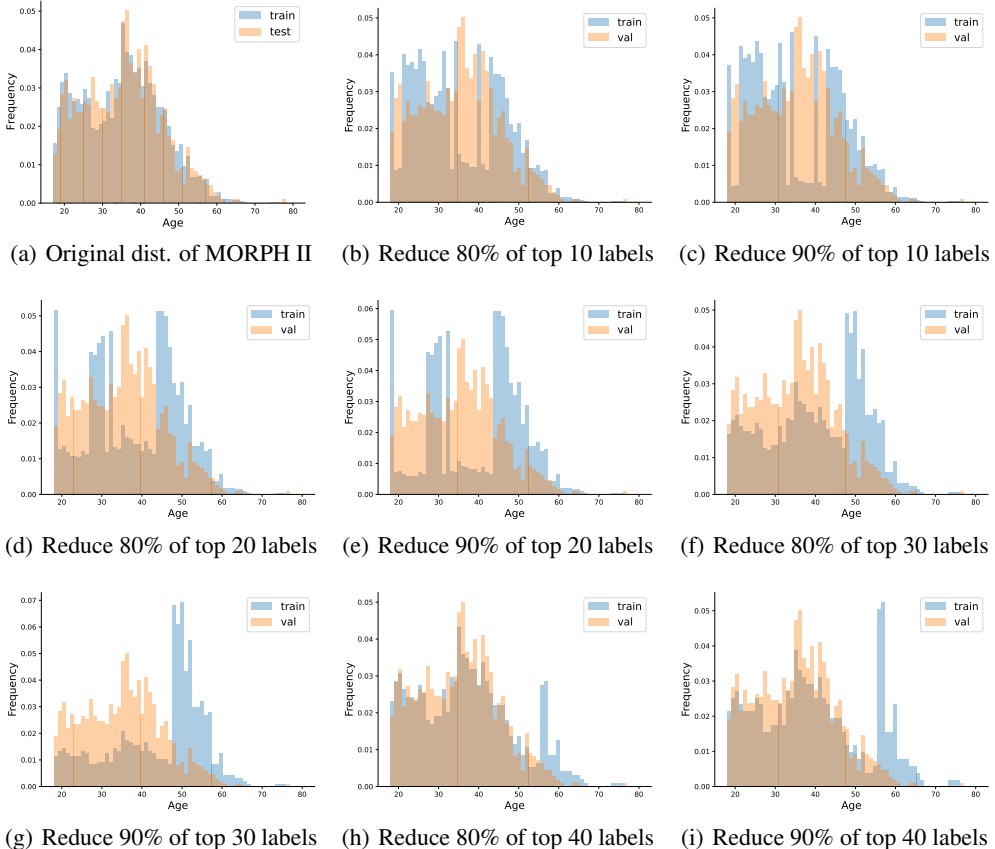

(a) Original dist. of MORPH II   (b) Reduce 80% of top 10 labels   (c) Reduce 90% of top 10 labels

(d) Reduce 80% of top 20 labels   (e) Reduce 90% of top 20 labels   (f) Reduce 80% of top 30 labels

(g) Reduce 90% of top 30 labels   (h) Reduce 80% of top 40 labels   (i) Reduce 90% of top 40 labels

Figure 1: Original and shifted distributions of the MORPH II dataset.

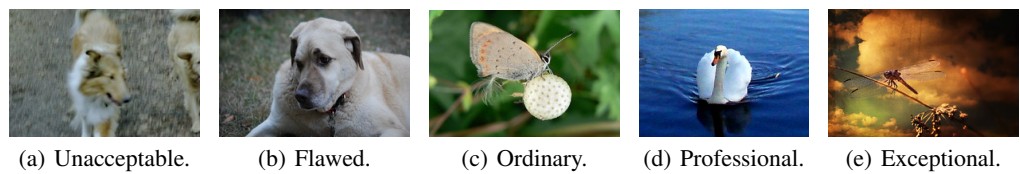

(a) Unacceptable.   (b) Flawed.   (c) Ordinary.   (d) Professional.   (e) Exceptional.

Figure 2: Samples from the animal collections of the Aesthetics dataset.

We list quantitative (Table 1) and qualitative (Table 4) results for CNN Baseline, CLIP without context initialization, and CLIP with context initialization. We observe that the prototypes for CNN are quite sparse and unordered. Almost half of the prototypes violate the ordinal property for each model. The initialized context helps ordinality and can further boost the performance of both CoOp [12] and OrdinalCLIP. Note that both models start to be optimized from the above two CLIP-produced language prototypes. Both models can improve the ordinality with their learned prototypes.

Table 1: The ordinality scores of language prototypes for CNN baseline, CLIP without initialized context embeddings, and CLIP with initialized context embeddings.

| Model | CNN Baseline | CLIP Rand. Ctx | CLIP Init. Ctx |
|---|---|---|---|
| Ordinality (%) | 49.13 | 49.71 | 54.84 |
| MAE | 2.63 | 19.75 | 14.45 |

Table 2 and Table 3 show the corresponding ordinality scores for the ablation study. By incorporating initialized context embeddings, the ordinality and performance can be further improved. Under all prompt optimization strategies, OrdinalCLIP is constantly better than CoOp [12]. Note that by

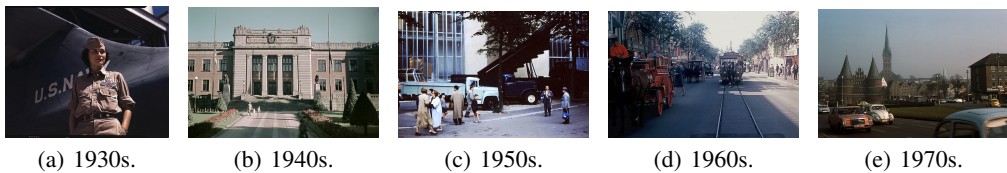

(a) 1930s.      (b) 1940s.      (c) 1950s.      (d) 1960s.      (e) 1970s.

Figure 3: Samples from the historical image dating dataset.

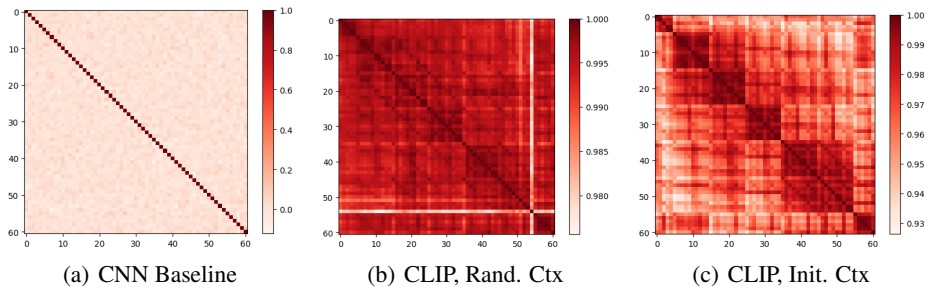

(a) CNN Baseline      (b) CLIP, Rand. Ctx      (c) CLIP, Init. Ctx

Figure 4: The similarity matrices of language prototypes for CNN baseline, CLIP without initialized context embeddings, and CLIP with initialized context embeddings.

tuning both context and rank embeddings, the ordinality of both models is decreased yet the MAE is improved. We attribute this to the different optimization difficulties and different capacities of the prompt learner. Figure 5 and Figure 6 visualize the similarity matrices for each setting. We can see that OrdinalCLIP learns smoother language prototypes with increased ordinality.

Table 2: CoOp Ablation with ordinality on the MORPH II dataset.

| Model | CoOp [12] | | | | | |
|---|---|---|---|---|---|---|
| Tune Rank | ✓ | ✓ | | | ✓ | ✓ |
| Tune Ctx | | | ✓ | ✓ | ✓ | ✓ |
| Init Ctx | | ✓ | | ✓ | | ✓ |
| Ordinality (%) | 52.67 | 55.68 | 55.95 | 59.92 | 54.52 | 55.00 |
| MAE | 2.58 | 2.41 | 2.44 | 2.39 | 2.39 | 2.39 |

Table 4 presents the ordinality under few-shot settings. For each few-shot setting, the ordinality of CNN is not changed due to its sparse prototypes. Figure 7 further illustrates such effect. OrdinalCLIP can obtain higher ordinality compared with CoOp [12]. With more data involved, the ordinality of both models is increased. Figure 8 and Figure 9 present their similarity matrices respectively. We find that OrdinalCLIP can learn more compact and smoother language prototypes.

Table 5 and Figure 10 11 12 illustrate the quantitative and qualitative results under distribution shift settings. Similar observations as those under few-shot settings are found.

# D  The Visualization of Interpolation Weight Matrix

The toy visualization of the interpolation weight matrix is shown in Figure 13. This toy example consists of 17 ranks (in a row) and 10 base ranks (in a column). Each row indicates that how a rank embeddings is interpolated via the base ones. Two interpolation methods inject different ordinal properties into the rank embeddings. We observe that linear interpolation gives smoother weights and inverse property interpolation gives sharper weights

Table 3: OrdinalCLIP Ablation with ordinality on the MORPH II dataset.

| Model | OrdinalCLIP | | | | | |
|---|---|---|---|---|---|---|
| Tune Rank | ✓ | ✓ | | | ✓ | ✓ |
| Tune Ctx | | | ✓ | ✓ | ✓ | ✓ |
| Init Ctx | | ✓ | | ✓ | | ✓ |
| Ordinality (%) | 73.66 | 77.10 | 74.56 | 80.86 | 65.20 | 66.63 |
| MAE | 2.38 | 2.34 | 2.37 | 2.34 | 2.36 | 2.32 |

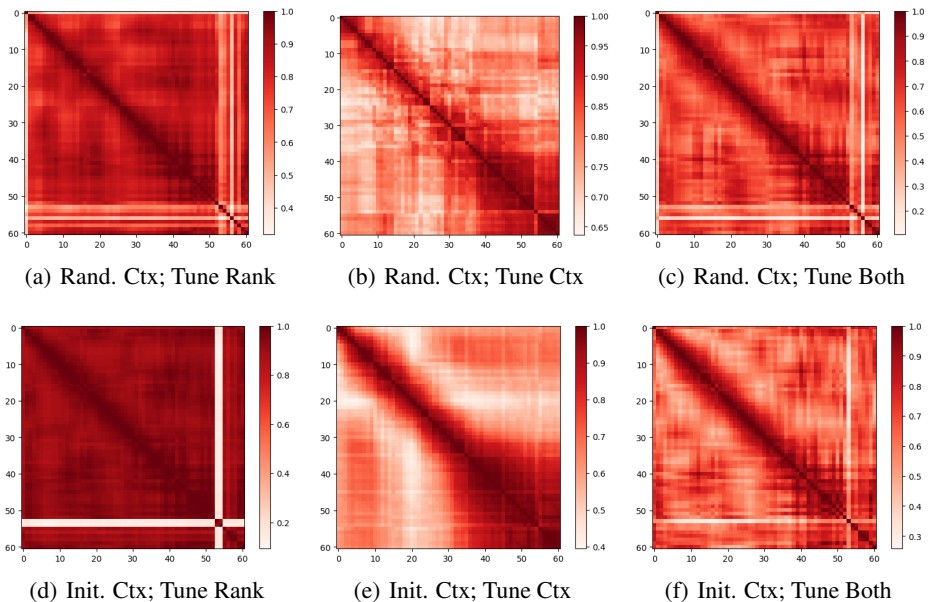

(a) Rand. Ctx; Tune Rank      (b) Rand. Ctx; Tune Ctx      (c) Rand. Ctx; Tune Both

(d) Init. Ctx; Tune Rank      (e) Init. Ctx; Tune Ctx      (f) Init. Ctx; Tune Both

Figure 5: The similarity matrices of language prototypes for CoOp [12].

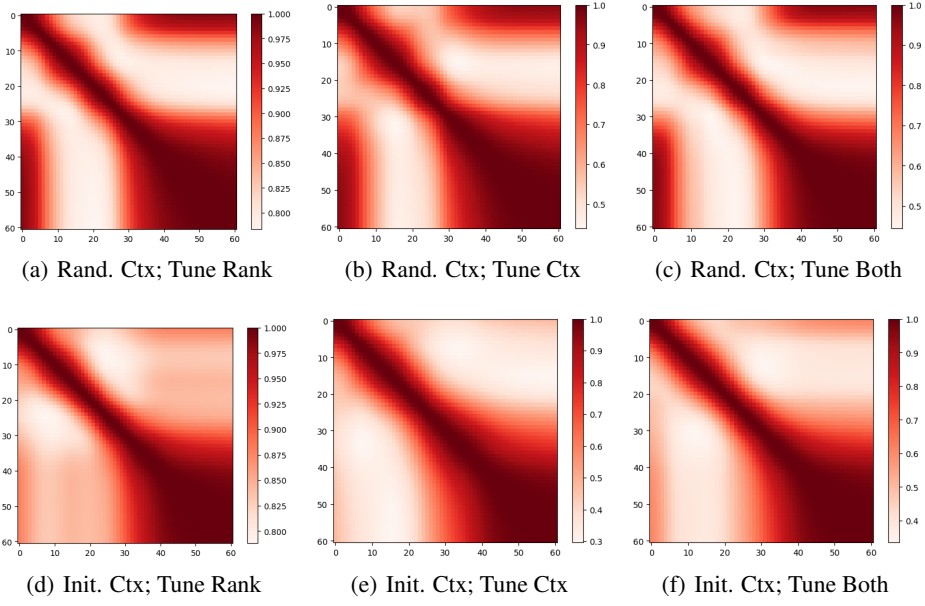

(a) Rand. Ctx; Tune Rank      (b) Rand. Ctx; Tune Ctx      (c) Rand. Ctx; Tune Both

(d) Init. Ctx; Tune Rank      (e) Init. Ctx; Tune Ctx      (f) Init. Ctx; Tune Both

Figure 6: The similarity matrices of language prototypes for OrdinalCLIP.

Table 4: We report the ordinality (%) results under few-shot settings on the MOPRH II dataset.

| #-Shots | 1 | 2 | 4 | 8 | 16 | 32 | 64 |
|---|---|---|---|---|---|---|---|
| CNN Baseline | 49.13 | 49.13 | 49.13 | 49.13 | 49.13 | 49.13 | 49.13 |
| CoOp [12] | 53.52 | 51.82 | 52.09 | 53.46 | 56.53 | 59.28 | 59.55 |
| OrdinalCLIP | 96.77 | 96.77 | 96.77 | 96.77 | 87.78 | 88.37 | 69.65 |

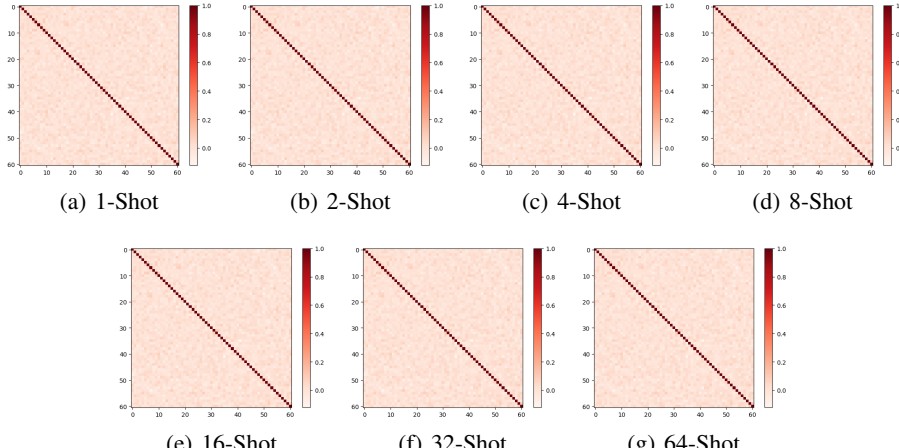

(a) 1-Shot    (b) 2-Shot    (c) 4-Shot    (d) 8-Shot

(e) 16-Shot    (f) 32-Shot    (g) 64-Shot

Figure 7: The similarity matrices of language prototypes for CNN Baseline on the few-shot task.

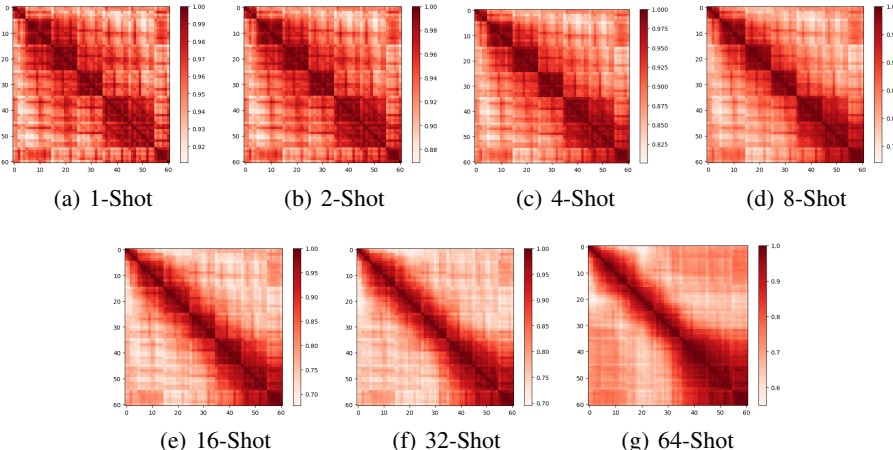

(a) 1-Shot    (b) 2-Shot    (c) 4-Shot    (d) 8-Shot

(e) 16-Shot    (f) 32-Shot    (g) 64-Shot

Figure 8: The similarity matrices of language prototypes for CoOp [12] on the few-shot task.

Table 5: The ordinality (%) results under the distribution shift setting on the MOPRH II. "re cls" denotes the number of reduced classes, and "re smp" means the percentage of reduced sampled in one class.

| re cls - re smp | 10-80 | 10-90 | 20-80 | 20-90 | 30-80 | 30-90 | 40-80 | 40-90 |
|---|---|---|---|---|---|---|---|---|
| CNN Baseline | 49.13 | 49.13 | 49.13 | 49.13 | 49.13 | 49.13 | 49.13 | 49.13 |
| CoOp [12] | 59.07 | 58.59 | 60.34 | 56.95 | 58.59 | 58.06 | 57.75 | 55.47 |
| OrdinalCLIP | 63.83 | 64.36 | 69.86 | 71.87 | 91.54 | 94.39 | 84.66 | 90.85 |

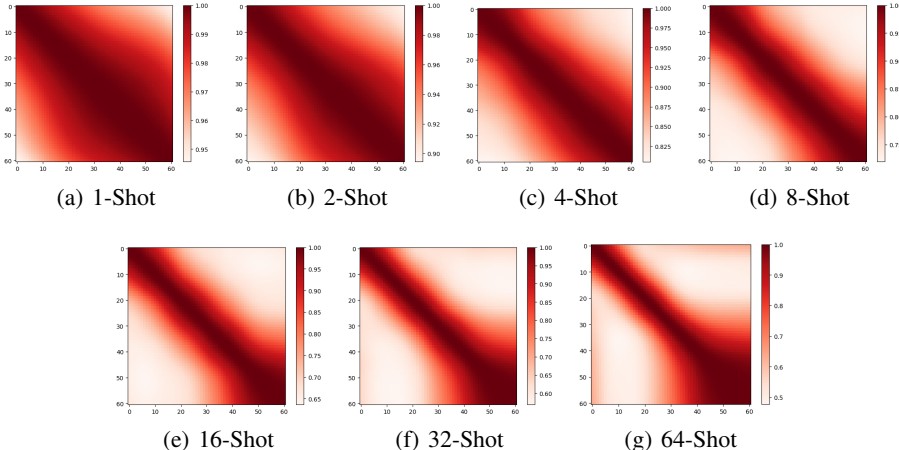

Figure 9: The similarity matrices of language prototypes for OrdinalCLIP on the few-shot task.

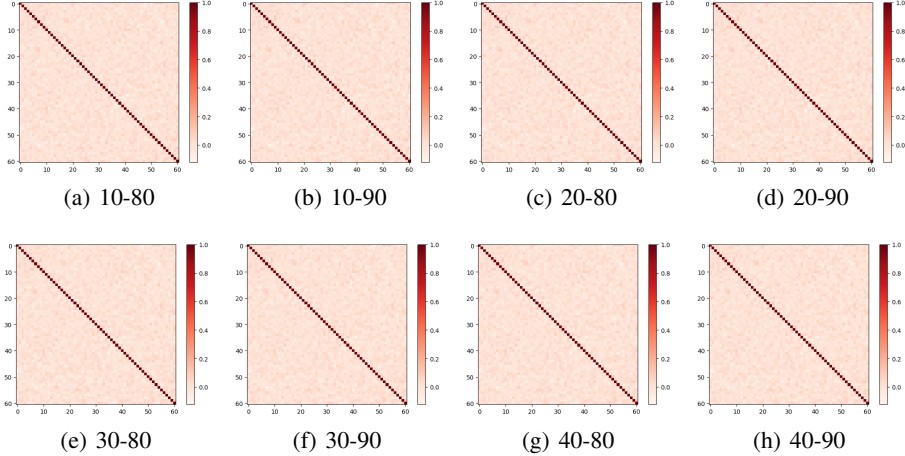

Figure 10: The similarity matrices of language prototypes for CNN Baseline on the distribution shift task. The first integer is "re cls", denoting the number of reduced classes; the second integer is "re smp", indicating the percentage of reduced sampled in one class

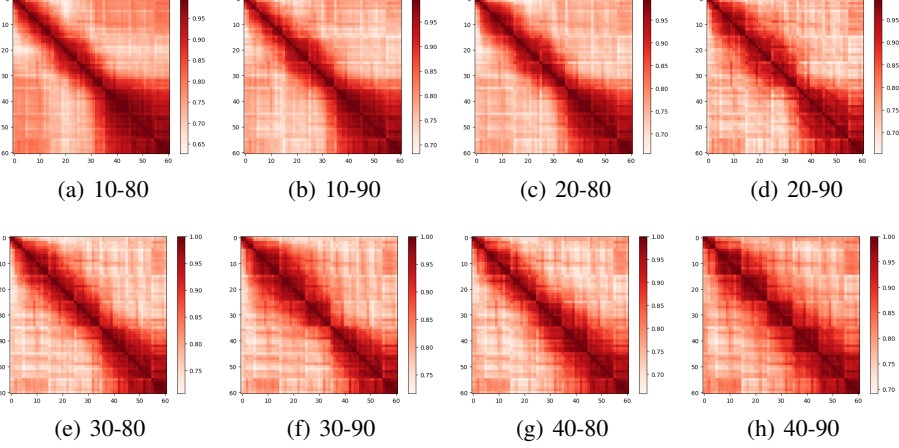

Figure 11: The similarity matrices of language prototypes for CoOp [12] on the distribution shift task. The first integer is "re cls", denoting the number of reduced classes; the second integer is "re smp", denoting the percentage of reduced sampled in one class

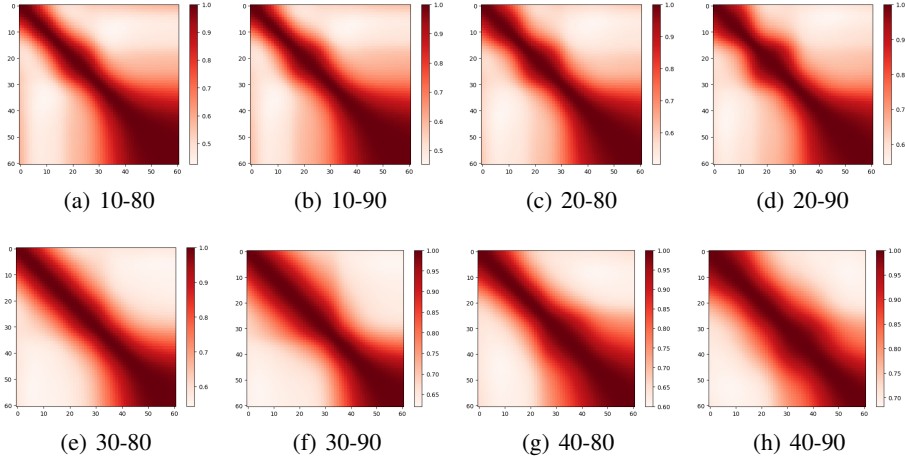

(a) 10-80       (b) 10-90       (c) 20-80       (d) 20-90

(e) 30-80       (f) 30-90       (g) 40-80       (h) 40-90

Figure 12: The similarity matrices of language prototypes for OrdinalCLIP on the distribution shift task. The first integer is "re cls", denoting the number of reduced classes; the second integer is "re smp", denoting the percentage of reduced sampled in one class

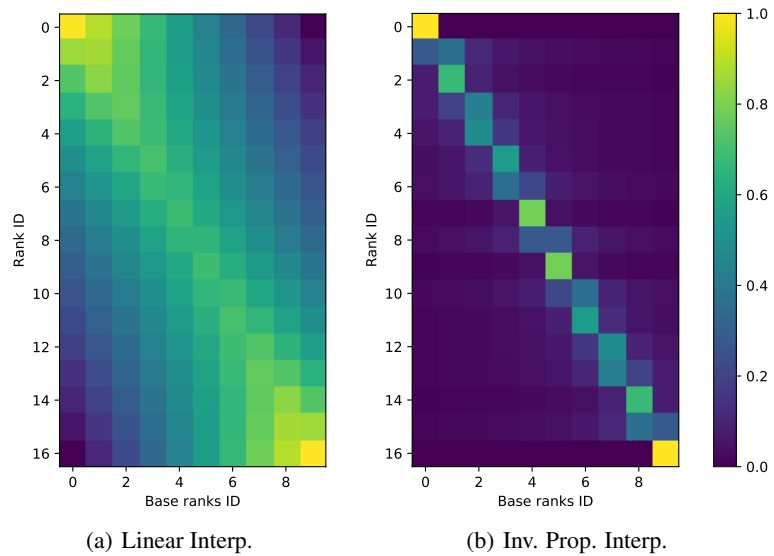

(a) Linear Interp.       (b) Inv. Prop. Interp.

Figure 13: The visualization of both linear and inverse-property interpolation.

# E  Liner Probe Experiment

We conducted experiments with the Linear probe solution on all tasks. The results are presented in Table 6. We see that our method consistently outperforms the Linear probe method on all datasets, which demonstrates the effectiveness of our method. It is worth pointing out that since most SOTA methods use VGG-16 as the vision encoder, we simply follow this setting for a fair comparison. Moreover, the specific choice of vision encoder does not affect our method and conclusion.

Table 6: The results of zero-shot with or without Language Prior (LP) on MORPH II. We give two settings, random initialized language prototypes (without LP) and language initialized prototypes (with LP). The features of images are extracted by the fixed CLIP image encoder. The significant better performance of CLIP zero-shot indicates that the language prior contain certain level of ordinal information.

| Dataset | MAE - lower is better | | Acc. - higher is better  (%) | |
| --- | --- | --- | --- | --- |
| | Linear Probe | OdrinalCLIP | Linear Probe | OdrinalCLIP |
| MORPH II | 4.70 | **2.32** | - | - |
| Adience | 0.64 | **0.47** | 51.80 | **61.20** |
| Aesthetics | 0.487 | **0.280** | 61.60 | **72.85** |
| Historical | 0.86 | **0.67** | 41.07 | **56.44** |