# OpenReview forum: "OrdinalCLIP: Learning Rank Prompts for Language-Guided Ordinal Regression"
_NeurIPS.cc/2022/Conference — NeurIPS 2022 Accept_

### Official Review · Reviewer_RX3e · 2022-07-10

**Rating:** 6
**Confidence:** 4
**Soundness:** 3 good
**Presentation:** 2 fair
**Contribution:** 3 good

**Summary:**

The authors propose a language-powered model for ordinal regression based on CLIP. The language prototypes are constructed from sentences with rank categories via the CLIP paper encoder, and then optimizing the CLIP model by language prototype and image feature matching. To further boost the ordinality, this paper introduces the learnable rank prompts by interpolation from the base rank embeddings. Multiple experiments on age estimation, image aesthetics assessment and historical image dating show that the proposed paradigm surpasses other related methods.

**Questions:**

1. I would suggest that the authors better explain the motivation of the paper and the description of the tasks presented. While the ordinal regression approach proposed by the authors is novel and makes sense, I did not see a description of the task or the challenges.
2. Since this paper uses CoOP as a competitive method for comparison, the paper lacks a description of how to implement CoOP into ordinal regression. Could the authors discuss this further?


**Limitations:**

The authors have discussed the limitations and potential negative effects of their work.

**Strengths And Weaknesses:**

Strengths:
1. This paper introduces the contrastive language-image pretrained (CLIP) model as a paradigm for ordinal regression is novel to me.
2. The proposed language prototypes and learnable rank prompts are insightful extensions of the CLIP model for the ordinal regression task.
3. The proposed interpolation learning rank prompts contribute to the output of smooth language prototype similarity trends, which represent well learned ordinality.

Weaknesses:
1. The writing of this paper should be improved , including motivation, related work and task description, etc.
2. The motivation for the two proposed interpolations, linear interpolation and inverse-proportion, is unclear and lacks visualization for comparison other than numerical comparison.

---

> ### Author Response · Authors · 2022-08-02
> **Response to the Reviewer RX3e (Part II)**
>
> **Q4: Description of Adapting CoOp into Ordinal Regression**
>
> > Since this paper uses CoOP as a competitive method for comparison, the paper lacks a description of how to implement CoOP into ordinal regression. Could the authors discuss this further?
>
> **[Reply]** Here we detail the implementation of CoOp [48] in the ordinal regression task. We borrow the CoOp model only with the modifications of language inputs. The prompt context (context embeddings) could be initialized by either task-related prompt templates (e.g, "The age of the person is {}." for age estimation) or random vectors. We change the input class label in CoOp to the rank labels of the task (e.g, "0", "1", ..., "99", "100", 100 ranks for age estimation). CoOp only finetunes the shared context embeddings ($m$ word embeddings). To fairly compare with OrdinalCLIP, we experiment with all three settings: only finetune the context embeddings, only finetune the rank embeddings, and finetune both context and rank embeddings.

---

> ### Author Response · Authors · 2022-08-02
> **Response to the Reviewer RX3e (Part I)**
>
> We thank Reviewer RX3e for the time, and also for the positive and constructive feedback! We are glad that the reviewer found our method to be **novel**, and the proposed language prototypes and learnable rank prompts to be **insightful extensions** of the CLIP model. We have revised the manuscript as suggested by the reviewer, and we address the reviewer's concerns as below.
>
> **Q1: Improve Writing**
>
> > The writing of this paper should be improved, including motivation, related work and task description, etc.
>
> **[Reply]** Many thanks for your suggestion. We have revised the related work section in our manuscript to be more organized. We also did our best to revise the introduction section of the manuscript to include a more detailed description of the tasks and challenges, as well as a clearer explanation of the motivation. We further elaborate on our modifications regarding the task description and motivation in the response to the next question.
>
>
> **Q2: Explain the motivation and the description of the tasks**
>
> > I would suggest that the authors better explain the motivation of the paper and the description of the tasks presented. While the ordinal regression approach proposed by the authors is novel and makes sense, I did not see a description of the task or the challenges.
>
> **[Reply]** Thanks for the advice. For a given image, the task of ordinal regression in computer vision is dedicated to predicting a rank number or continued value. For example, age estimation aims to estimate the age of a given face image while image aesthetic assessment predicts the aesthetic score for an image.
>
> As many popular methods adopt a classification framework, there are two main challenges.  First, treating ranks as independent class categories fails to grasp the ordinal property. Second, as the learned concepts are mainly derived from the training set, these approaches are prone to overfit and usually attain unsatisfactory performance.
>
> Since learning the rank concept from the image domain alone is prone to overfitting, we can leverage multimodal information to alleviate this issue. The human language contains rich semantic information and prior knowledge. We consider simultaneously borrowing the rank concept from the language domain. Specifically, each rank label is not only regarded as a class category but also linked to a sentence describing the corresponding rank, such as "this person is 23 years old". In this way, our model not only learns the concept of ranks defined on the vision dataset but also exploits the common knowledge of rank in the language domain. Therefore we propose a language-powered paradigm for ordinal regression to alleviate the overfitting issue by associating each rank category with its language concept. Moreover, we propose to learn rank prompts to model the ordinal property.
>
> We have included these explanations in the revised version.
>
>
>
> **Q3: Clarification of two proposed interpolations**
>
> > The motivation for the two proposed interpolations, linear interpolation, and inverse-proportion, is unclear and lacks visualization for comparison other than numerical comparison.
>
> **[Reply]** We have included an additional Figure [13] in Appendix, to visualize a toy example of the interpolation weight matrix. We use interpolation to incorporate the ordinal property from the language end. Specifically, the input word embeddings differ only one word embedding from each other. Then for each rank embedding, we need to incorporate a certain level of the ordinal property. In our implementation, we use linear interpolation and inverse property interpolation to impose the ordinality to the rank embeddings. Our experiments show that via interpolation between several base ranks, the language prototypes can better preserve the ordinality, resulting in a compact and ordinal latent space. In other words, the ordinality of the rank embeddings can be implicitly propagated toward the language prototypes. In practice, we consider two different interpolation strategies: linear interpolation and inverse property interpolation, where linear interpolation gives smoother weights and inverse property interpolation gives sharper weights. The experiments show that smoother weights usually give better results when the number of base ranks is small.

---

> ### Author Response · Authors · 2022-08-08
> **Looking forward to your feedback**
>
> Dear Reviewer RX3e,
>
> Thanks again for your valuable advice and supportive comments! We have responded to your initial comments. We are looking forward to your feedback and will be happy to answer any further questions you may have.

---

### Official Review · Reviewer_PfAX · 2022-07-10

**Rating:** 6
**Confidence:** 3
**Soundness:** 4 excellent
**Presentation:** 3 good
**Contribution:** 4 excellent

**Summary:**

The authors propose a language-powered paradigm for ordinal regression tasks by learning rank prompts, named OrdinalCLIP. The OrdinalCLIP can leverage rank categories of language to explicit learning ordinal rank embeddings, which will preserve the order of the language prototypes in the language latent space. In the three regression tasks, including age estimation, historical image dating, and image aesthetics assessment, The experimental results show good performance than other baseline models. In addition, for few-shot learning, the method also gains improvement. The overall structure is well-organised. The paper has a clear motivation and is innovative for the regression field.

**Questions:**

Questions:

1.To leverage the language priors with the text encoder, we treat the rank categories as words.  How to choose a suitable sentence? The sentence of “a person at the age of [rj] is the best? Are there some ablation studies?

2. we directly learn m word embeddings. what's the meaning of m? It's the ordinal number word?

3. See the weakness.


**Limitations:**

Yes

**Strengths And Weaknesses:**

Strength:

1. The innovative language-powered paradigm for ordinal regression uses language prototypes and learned rank prompts, which are interesting and valuable.

2. The good performance shows the effectiveness of OrdinalCLIP.

3. The approvement and experiments of the appendix are detailed.

Weakness:

1. The statement of introduction, related works, and problem statements should narrow down the ordinal regression to the vision-language or CV ordinal regression task cuz there are some pure language ordinal regression tasks.

2.The two loss (image-to-text loss and a text-to-image loss ) should be introduced in detailed, and the reason for using KL.

3. we choose to maintain the order of rank embeddings to preserve the order of the language prototypes. This statement is unclear. How to maintain the order of the languag?

---

> ### Author Response · Authors · 2022-08-02
> **Response to the Reviewer PfAX**
>
> We thank the reviewer for the constructive feedback and a positive assessment of our work. We are happy the reviewer finds the paper **well-organised and clear-motivated**, our method **interesting, valuable, and innovative with good performance**. Below we detail our answer to the review concerns.
>
> **Q1: Narrow down the writing to CV tasks**
>
> > The statement of introduction, related works, and problem statements should narrow down the ordinal regression to the vision-language or CV ordinal regression task cuz there are some pure language ordinal regression tasks.
>
> **[Reply]** Thanks for this nice suggestion! We have revised the introduction, related works, and problem statements of the manuscript to ensure narrowing down to ordinal regression tasks in computer vision.
>
> **Q2: Explanation of KL-Loss**
>
> > The two loss (image-to-text loss and a text-to-image loss ) should be introduced in detailed, and the reason for using KL.
>
> **[Reply]** Following CLIP, we use an image-to-text loss and a text-to-image loss to supervise the model. For image-to-text loss, using KL loss is equivalent to using cross-entropy loss, as the labels for each image are all one-hot encoded. However, for the text-to-image loss, there might be several image hits for a certain label in a mini-batch. We follow ActionCLIP [44] to use a KL loss to supervise the text-to-image logits. Specifically, the ground-truth matrix is constructed by taking the normalized probability of each multi-hot label for the corresponding rank.
>
> **Q3: Explanation of order preservation of rank embeddings and language prototypes**
>
> > We choose to maintain the order of rank embeddings to preserve the order of the language prototypes. This statement is unclear. How to maintain the order of the language?
>
> **[Reply]** Sorry for the confusion. We hope that the language prototypes will lie on the manifold in good order. Since the language prototypes are extracted from the CLIP model using prompt inputs, we instead consider constraining the prompt inputs. The inputs of text encoders are context embeddings ($m$ words) along with a rank embedding. The context embeddings are shared among all ranks. The input word embeddings differ only one word embedding from each other. Then for each rank embedding, we need to incorporate a certain level of the ordinal property. In our implementation, we use linear interpolation and inverse property interpolation to impose the ordinality to the rank embeddings. Our experiments show that via interpolation between ranks, the language prototypes can better preserve the ordinality, resulting in a compact and ordinal latent space. In other words, the ordinality of the rank embeddings can be implicitly propagated toward the language prototypes.
>
> **Q4: Ablation of different prompt templates:**
>
> > To leverage the language priors with the text encoder, we treat the rank categories as words. How to choose a suitable sentence? The sentence of “a person at the age of [rj] is the best? Are there some ablation studies?
>
> **[Reply]** The prompt templates for ablation are shown in the tables below.
>
> | Ctx. Ind. | Template Ctx.                                                 |
> |:---------:|---------------------------------------------------------------|
> |    0-0    | Age estimation: the age of the person is {} .                 |
> |    1-0    | Age estimation: the age of the person in the portrait is {} . |
> |    2-0    | Age estimation: the age is {} .                               |
> |    3-0    | Age estimation: the age of the face is {} .                   |
> |    0-1    | The age of the person is {} .                                 |
> |    1-1    | The age of the person in the portrait is {} .                 |
> |    2-1    | The age is {} .                                               |
> |    3-1    | The age of the face is {} .                                   |
>
> The table below shows that different optimization start points all lead to similar convergence and performance, which suggests that the most meaningful templates work fine for this task.
>
> | Ctx. Ind.   |  0-0 |  1-0 |  2-0 |  3-0 |  0-1 |  1-1 |  2-1 |  3-1 |
> |-------------|:----:|:----:|:----:|:----:|:----:|:----:|:----:|:----:|
> | OrdinalCLIP | 2.30 | 2.31 | 2.30 | 2.32 | 2.31 | 2.32 | 2.32 | 2.31 |
>
> **Q5: Explanation of "$m$ word embeddings"**
>
> > We directly learn m word embeddings. what's the meaning of m? It's the ordinal number word?
>
> **[Reply]** $m$ word embeddings refer to the shared context embeddings for all ranks, where $m$ is the number of the context embeddings. For example, given a prompt context "This person is {}.", it has $m=3$ word embeddings (in fact, it is $m=4$, as the dot "." is also a word embedding in CLIP tokenizer).

---

> ### Author Response · Authors · 2022-08-08
> **Looking forward to your feedback**
>
> Dear Reviewer PfAX,
>
> Thanks again for your valuable advice and supportive comments! We have responded to your initial comments. We are looking forward to your feedback and will be happy to answer any further questions you may have.

---

### Official Review · Reviewer_wMGt · 2022-07-11

**Rating:** 5
**Confidence:** 4
**Soundness:** 2 fair
**Presentation:** 2 fair
**Contribution:** 2 fair

**Summary:**

The paper proposes a method to use CLIP for ordinal regression using a combination of soft labels and prefix tuning, with an interpolation scheme added to enforce order between the learnt prompts.

**Questions:**

Please see the strengths and weaknesses.

**Limitations:**

Limitations are not discussed.

**Strengths And Weaknesses:**

# Strengths
The idea of using context from natural language for ordinal regression tasks is interesting and worth exploring. The proposed interpolation method to allow regressing in between ordinal ranks is also clever.

# Weaknesses
The main weakness to me is that an obvious baseline is missing from an engineering perspective. The paper uses a VGG-16 network pretrained on ImageNet as a trainable vision encoder for several datasets. The VGG-16 network has 138M parameters, similar to the ViT in the official CLIP release. The natural baseline would then have been to train a linear probe atop the CLIP ViT-B to simply predict the rank as a classification task. Second, the results in Fig 3. are very surprising. If I am reading it correctly, about 35% of the rank prototypes violate the ordinal property. This is a substantial portion, and suggests the proposed method cannot grasp anything on the long tail of the distribution. Third, _why_ do language priors help with this task at all? It's difficult to conceive that CLIP has learned a meaningful representation of say, the number 72. Even if CLIP did have a very meaningful representation of some arbitrary number, the rank embeds are entirely learned. How is the language information being used?  T1 is missing variance numbers for the comparison with CoOp. As evident in T2, CoOp and OrdinalClip are very close, with large variances. The authors should include a statistical test to confirm that the proposed method is indeed superior to CoOp. There are no variance numbers again in Table 7.

# Summary
The technique presented is a somewhat straightforward extension of CoOp, with the main novelty being that the literal rank names are replaced with soft class names. The results are in general very close to that of CoOp, and some tables are missing variance information, while others have variance information. Given how close the results of the method are to CoOp and how high the included variances are, variance results should be included for all tables, and statistical significance tests conducted. An important baseline (linear probing) is missing. Furthermore, it is unclear to me why language information can help with this kind of task at all, I do not see any experiments to explain why language information can help with this kind of task, or provide insight into what the language model is adding here. Finally, the result showing that 35% of the rank embeddings do not obey the ordinal property is troubling, especially since the "broken" rank embeddings are clustered in the long tail. This suggests the error distribution of the model is highly biased and the model does not work well at all for the tail.

---

> ### Author Response · Authors · 2022-08-02
> **Response to the Reviewer wMGt (Part III)**
>
> **Q5: Variance and Statistical Tests:**
> > T1 is missing variance numbers for the comparison with CoOp. As evident in T2, CoOp and OrdinalClip are very close, with large variances. The authors should include a statistical test to confirm that the proposed method is indeed superior to CoOp. There are no variance numbers again in Table 7.
>
> > The results are in general very close to that of CoOp, and some tables are missing variance information, while others have variance information. Given how close the results of the method are to CoOp and how high the included variances are, variance results should be included for all tables, and statistical significance tests conducted.
>
> **[Reply]** We believe there is a huge misunderstanding here. The variances in Tables 2 and 8 are the variances of the five-fold cross-validation results, which means that they **do not represent** the performance stability of the model in **multiple runs**, but represent the performance stability of the model **among the five-fold sub-datasets**. The high variance simply means the difference among the five-fold sub-datasets.
>
>
> We did not report the variance results in Tables 1 and 7 simply following the practice of previous SOTA methods [6,20,22,41,45]. Following the advice, we provide the variance results and statistical significance tests for all tasks.
>
>
> For MORPH II, the standard deviation results over five random runs are shown below. We see they are quite small. A one-way ANOVA test revealed a significant performance boost of OrdinaCLIP [$\alpha=0.05, F(2, 12)<240.7$, $p=2e{-10}$]. *Post-hoc* analyses (Tukey HSD) showed that the results of OrdinalCLIP are significantly better than those from CoOp and those from CNN baseline.
>
>
> **Table R1-7. The standard deviation results over five random runs on the MORPH II database.**
> Method | CNN Baseline |  CoOp |  OrdinalCLIP
> ---|:---:|:---:|:---:|
> Standard Deviation | 0.04 | 0.02 | 0.01
>
>
> For Adience dataset, the paired t-test [$\alpha=0.05, P(T<=t)=0.020<0.05$ for Acc; $\alpha=0.05, P(T<=t)=0.002<0.05$ for MAE] reveals the significant performance of OrdinalCLIP over CoOp.
>
> For the Image Aesthetics task, we show the standard deviation of the five-fold cross-validation results below. Paired t-test on both MAE and Accuracy metrics [$\alpha=0.05, P(T<=t)=0.026<0.05$ for Acc; $\alpha=0.05, (T<=t)=0.018<0.05$ for MAE] reveal the significant performance improvement made by OrdinalCLIP, compared with CoOp.
>
> **Table R1-8. The standard deviation results for the Accuracy metric on the Image Aesthetics dataset.**
> Method | Nature |  Animal |  Urban  | People |  Overall
> ---|:---:|:---:|:---:|:---:|:---:
> CNN Baseline | 1.84| 2.60 |1.83 |2.60 |2.22
> Zero-shot CLIP | 1.12| 2.33 | 2.20 | 1.57 | 1.80
> CoOp | 1.97 | 1.97 | 3.27 | 1.80 | 2.25
> OrdinalCLIP | 2.45 | 1.31 | 1.80 | 1.96 | 1.88
>
> **Table R1-9. The standard deviation results for the MAE metric on the Image Aesthetics dataset.**
> Method | Nature |  Animal |  Urban  | People |  Overall
> ---|:---:|:---:|:---:|:---:|:---:
> CNN Baseline | 0.017| 0.020 | 0.017 | 0.017 | 0.018
> Zero-shot CLIP | 0.007| 0.019 | 0.034 | 0.022 | 0.020
> CoOp | 0.020 | 0.023 | 0.033 | 0.004 | 0.020
> OrdinalCLIP | 0.027 | 0.016 | 0.018 | 0.021 | 0.020
>
> We also conducted the paired t-test on the Historical Dating dataset. The test [$\alpha=0.05, P(T<=t)=0.002<0.05$ for Acc; $\alpha=0.05, P(T<=t)=0.009<0.05$ for MAE] shows OrdinalCLIP is significant better than CoOp.
>
> We see that our method significantly outperforms CoOp and passes statistical significance tests on all tasks.
>
>
> **Q6: Novelty**
>
> >The technique presented is a somewhat straightforward extension of CoOp, with the main novelty being that the literal rank names are replaced with soft class names.
>
> **[Reply]** We do not agree. Simply adapting CoOp with learnable rank embeddings leads to no performance boost and degradation in ordinality (Table [10] and Figure [8] in Appendix). Our method is the first language-powered paradigm for ordinal regression. We first reformulate the ordinal regression task as the image-text matching to utilize the well-structured, rich-semantic CLIP latent space. To improve both performance and ordinality, we propose to construct the ordinal rank embeddings via interpolation between a set of base rank embeddings that are learnable during training. These are specifically designed for ordinal regression tasks  and are non-trivial. Moreover, both reviewers PfAX and RX3e appreciated our work and agreed that our method was novel (reviewer RX3e also agreed that our method was ''*insightful extensions of the CLIP model for the ordinal regression task*'').

---

> ### Author Response · Authors · 2022-08-02
> **Response to the Reviewer wMGt (Part II)**
>
> **Q3: Long tail issue**
>
> >the proposed method cannot grasp anything on the long tail of the distribution.
>
> >rank embeddings are clustered in the long tail ... suggests the error distribution of the model is highly biased and the model does not work well at all for the tail.
>
> **[Reply]** First, as analyzed above, our approach **does not have a substantial portion** of language prototypes that violate the ordinal property. Second, Learning from long tail data is difficult for all methods but **our method still outperforms other methods on long tail data**.
>
> We define the data in the last 25 ranks on the MORPH II dataset as long tail data (the data distribution is shown in the supplementary material). We first present the ordinality score for the last 25 ranks on the MORPH II database. We see our method achieves much higher ordinality scores.
>
> **Table R1-4. The Ordinality@s results for the last 25 ranks on the MORPH II database. Ordinality@$\infty$ denotes the old metric.**
> Ordinality | @2 | @4 | @8 | @16 |  @$\infty$
> ---|:---:|:---:|:---:|:---:|:---:
> CNN Baseline | 70.21% | 58.89% | 52.44% | 51.89% | 51.00%
> CoOp [48] | 89.36% | 74.44% | 65.24% | 62.50% | 60.33%
> OrdinalCLIP | **100.00%** | **100.00%** | **100.00%** | **97.64%** | **97.14%**
>
> We further show the MAE results on the long tail data, which also demonstrates that our method can better handle the long tail data.
>
> **Table R1-5. The MAE results on the long tail data of MORPH II database.**
> Method | CNN Baseline | CoOp [48] | OrdinalCLIP
> ---|:---:|:---:|:---:
> MAE  | 4.32 | 4.21 | **4.06**
>
> **Q4: Why and how does language information help with this task**
>
> > why do language priors help with this task at all? It's difficult to conceive that CLIP has learned a meaningful representation of say, the number 72. Even if CLIP did have a very meaningful representation of some arbitrary number, the rank embeds are entirely learned. How is the language information being used?
>
> > Furthermore, it is unclear to me why language information can help with this kind of task at all, I do not see any experiments to explain why language information can help with this kind of task, or provide insight into what the language model is adding here.
>
>
> **[Reply]** Why does language information help with this task? Existing methods are easy to overfit and usually attain unsatisfactory performance as the learned rank concepts are mainly derived from the vision training set. Since learning the rank concept from the image domain alone is prone to overfitting, we can leverage multimodal information to alleviate this issue. The human language contains rich semantic information and prior knowledge. We consider simultaneously borrowing the rank concept from the language domain. Specifically, each rank label is not only regarded as a class category but also linked to a sentence describing the corresponding rank, such as "this person is 23 years old". In this way, our model not only learns the concept of ranks defined on the vision dataset but also exploits the common knowledge of rank in the language domain.
>
> How does language information help with this task? In practice, we employ the pre-trained giant text encoder in CLIP to extract language prototypes for all ranks. Since the prototypes are obtained from a ***fixed language model***, we are somehow distilling the language knowledge from the CLIP model. Moreover, the prototypes are constrained in the well-learned language latent space, which is also a kind of regularization leading to stronger generalization.
>
> Any experiments? To see the benefits of language priors, we first consider the zero-shot setting. We conducted two experiments: 1) without Language Priors (w/o LP), the classifier is a random initialized FC layer, 2) with Language Priors (w/ LP), the classifier is language initialized prototypes with the CLIP text encoder. Neither experiment involves model training. The results in Table R1-6 show that the w/ LP solution significantly outperforms the w/o LP across four datasets, which indicates that the CLIP model does contain a meaningful representation of rank numbers to some extent, and language information can help with this task.
>
> We agree that CLIP may not be able to give a perfect representation of some arbitrary number **simply using raw text input**. Therefore we propose to learn rank prompts. Here we consider the full-training setting, where the full model is trained. w/ P refers to our OrdinalCLIP and w/o LP means that the language prototypes are replaced with an FC layer. The results show the effectiveness of language priors again.
>
> **Table R1-6. The MAE results on four benchmarks. The lower, the better.**
> Dataset | MORPH II  | Adience |  Image Aesthetics  | Historical Image Dating
> ---|:---:|:---:|:---:|---
> zero-shot, w/o LP | 32.51 | 2.73 | 1.425 | 1.95
> zero-shot, w/ LP | 14.45 | 1.50 | 0.730 | 1.48
> full training, w/o LP | 2.63 | 0.56 | 0.365 | 0.80
> full training, w/ LP | 2.32 | 0.47 | 0.280 | 0.67

---

> ### Author Response · Authors · 2022-08-02
> **Response to the Reviewer wMGt (Part I)**
>
> We would like to thank the reviewer for the valuable comments.  However, we feel there is some misunderstanding. We clarify the issues and address the questions accordingly as described below.
>
> **Q1: Linear probe baseline**
>
> > The main weakness to me is that an obvious baseline is missing from an engineering perspective. The paper uses a VGG-16 network pretrained on ImageNet as a trainable vision encoder for several datasets. The VGG-16 network has 138M parameters, similar to the ViT in the official CLIP release. The natural baseline would then have been to train a linear probe atop the CLIP ViT-B to simply predict the rank as a classification task.
>
> > An important baseline (linear probing) is missing.
>
> **[Reply]** Thanks for the advice. Following the suggestion, we conducted experiments with the Linear probe solution on all tasks. The results are presented below.
>
>
> **Table R1-1. The MAE results on four benchmarks. The lower, the better.**
> Dataset | MORPH II  | Adience |  Image Aesthetics  | Historical Image Dating
> ---|:---:|:---:|:---:|---
> Linear probe | 4.70 | 0.64 | 0.487 | 0.86
> OdrinalCLIP | **2.32** | **0.47** | **0.280** | **0.67**
>
> **Table R1-2. The Accuracy results on three benchmarks. The higher, the better.**
> Dataset | Adience |  Image Aesthetics  | Historical Image Dating
> ---|:---:|:---:|:---:
> Linear probe |  51.8% | 61.60% | 41.07%
> OdrinalCLIP |  **61.2%** | **72.85%**| **56.44%**
>
> We see that our method consistently outperforms the Linear probe method on all datasets, which demonstrates the effectiveness of our method. We have also included these results in our revision. It is worth pointing out that since most SOTA methods use VGG-16 as the vision encoder, we simply follow this setting for a fair comparison. Moreover, the specific choice of vision encoder does not affect our method and conclusion.
>
> **Q2: Definition and Effectiveness of Ordinality**
>
> > Second, the results in Fig 3. are very surprising. If I am reading it correctly, about 35% of the rank prototypes violate the ordinal property. This is a substantial portion...
>
> >  Finally, the result showing that 35% of the rank embeddings do not obey the ordinal property is troubling...
>
> **[Reply]** First, we clarify the misunderstanding of the ordinality definition. To evaluate the order of the learned language prototypes, we proposed a simple metric (line 235, footnote) that compares the normalized cosine distances of **every pair** of language prototypes.  The overall comparison is $(N - 1) \times N / 2$ given $N$ language prototypes. In Figure 3, there are around 35% of prototype **pairs** that violate the ordinal property, instead of 35% of prototypes.
>
> The second thing we want to clarify is that the ordinality score metric is only used to describe **relative ordinality** rather than **absolute ordinality**. The ordinality score is not a perfect metric as it assumes that the normalized cosine distances can be used to measure the order relations of language prototypes. Since the language prototypes lie in a nonlinear manifold, this assumption is difficult to hold between language priors that are far away. A better metric is to compute the language prototype pairs within a sliding window, where the assumption of a locally linear manifold can hold. We denote this metric as "Ordinality@s", where s is the sliding window size. The results on the MORPH II dataset show that our OrdinalCLIP can well preserve the order of language prototypes (100% for Ordinality@8).
>
> Thus, our approach **does not have a substantial portion** of language prototypes that violate the ordinal property.
>
> **Table R1-3. The Ordinality@s results on the MORPH II database.**
> Ordinality | @2 | @4 | @8 | @16
> ---|:---:|:---:|:---:|:---:
> CNN Baseline | 77.31% |  61.97% | 56.42% | 52.02%
> CoOp [48] |91.60% | 85.04% | 78.54% | 68.33%
> OrdinalCLIP | **100.00%** | **100.00%** | **100.00%** | **96.19%**

---

> ### Author Response · Authors · 2022-08-07
> **Looking forward to your feedback**
>
> Dear reviewer wMGt,
>
> Does our response address all of your concerns? As we have clarified some crucial misunderstandings about our paper, we sincerely hope that you can reconsider the rating. Please feel free to let us know if you have any further questions.

---

> > ### Comment · Reviewer_wMGt · 2022-08-07
> > **Response**
> >
> > Hello,
> >
> > The additional experiments show that the empirical performance of the method is very good. I will raise my rating.

---

> > > ### Author Response · Authors · 2022-08-08
> > > **Thank you**
> > >
> > > Thank you for your reply and your positive feedback. We appreciate it very much!

---

### Meta-Review · Area_Chair_dGL3 · 2022-08-31

**Recommendation:** Accept
**Confidence:** Certain

**Metareview:**

The paper proposes a language-powered model for ordinal regression tasks, based on CLIP. Language prototypes are constructed from sentences with rank categories via the CLIP paper encoder, and then optimizing the CLIP model by language prototype and image feature matching. To further boost the ordinality, this paper introduces the learnable rank prompts by interpolation from the base rank embeddings.

While the proposed approach builds on CoOp, reviewers agree the contribution is significant enough and original enough for NeurIPS. Regarding the experimental section, the paper shows that on three regression tasks (age estimation, historical image dating, and image aesthetics assessment), results show good performance compared to baseline models.

Concerns regarding the writing of the manuscript have been raised [PfAX, RX3e], but seem to have been addressed during the rebuttal phase.

**Award:**

No

---

### Decision · Program_Chairs · 2022-09-14

Accept